# SARS-CoV-2 Infection of the Central Nervous System: A Case Report

**DOI:** 10.3390/v16121962

**Published:** 2024-12-21

**Authors:** Trifon Valkov, Radka Argirova, George Dimitrov

**Affiliations:** 1Department of Infectious Diseases, Medical University of Sofia, Prof. Ivan Kirov hospital, 1431 Sofia, Bulgaria; t.valkov@medfac.mu-sofia.bg; 2Department of Clinical Laboratory, Acibadem City Clinic Tokuda Hospital, 1407 Sofia, Bulgaria; radkaargirova@abv.bg; 3Department of Medical Oncology, Medical University of Sofia, University Hospital “Tsaritsa Yoanna”, 1527 Sofia, Bulgaria

**Keywords:** SARS-CoV-2, neuroinfection, brain fog, long COVID

## Abstract

Central nervous system (CNS) infections caused by SARS-CoV-2 are uncommon. This case report describes the clinical progression of a 92-year-old female who developed a persistent neuroinfection associated with SARS-CoV-2. The patient initially presented with progressive fatigue, catarrhal symptoms, and a fever (38.6 °C). Initial laboratory findings revealed hypoxemia (O_2_ saturation 79.8%), acidosis (pH 7.3), an elevated C-reactive protein (CRP) level of 14.8 mg/L, and a high D-dimer level (2.15 µg/mL). Nasopharyngeal (NP) antigen and RT-PCR tests confirmed SARS-CoV-2 infection, and an NP swab also detected penicillin- and ampicillin-resistant Staphylococcus aureus. She was admitted for conservative management, including oxygen supplementation, IV fluids, and prophylactic anticoagulation. Subsequently, she developed neurological symptoms—lethargy, discoordination, and impaired communication—without signs of meningism. Cerebrospinal fluid (CSF) analysis identified SARS-CoV-2 RNA (Ct = 29) on RT-PCR, while bacterial cultures remained negative. Treatment was intensified to include 10% mannitol, dexamethasone, and empiric ceftriaxone. Despite these interventions, the patient remained somnolent, with a Glasgow Coma Scale (GCS) score of 10. Upon discharge, her GCS had improved to 14; however, she continued to experience lethargy and cognitive issues, commonly described as “brain fog”. Inflammatory markers remained elevated (CRP 23 mg/L) and repeat RT-PCR of CSF confirmed a persistent SARS-CoV-2 presence (Ct = 31). This case underscores the potential for SARS-CoV-2 to cause prolonged CNS involvement, leading to persistent neurological impairment despite standard therapy. Further research is essential to clarify the pathophysiology of and determine optimal management for SARS-CoV-2 neuroinfections.

## 1. Introduction

In December 2019, the novel coronavirus disease (COVID-19), caused by Severe Acute Respiratory Syndrome Coronavirus 2 (SARS-CoV-2), emerged in Hubei province, China, and quickly spread worldwide [1]. Like many respiratory infections, it primarily spreads through large droplets, breath aerosol, and contaminated surfaces [2]. While respiratory symptoms are the hallmark of COVID-19, neurological complications such as encephalopathy, encephalitis, cerebrovascular events, seizures, anosmia, and cognitive impairments have also been reported [3,4]. These neurological symptoms can occur during the acute phase, or persist for weeks to months, a condition often referred to as “long COVID” [5].

Research has demonstrated that SARS-CoV-2 can infect human neurons and astrocytes in vitro, leading to viral replication and neuronal damage [6]. The virus appears to enter neural cells via the endosomal pathway, which can be inhibited by blocking phosphoinositol-5 kinase, a host cell enzyme [6]. However, the exact mechanisms behind COVID-19-related neuroinfections remain debated. Some argue that these symptoms result from the direct viral invasion of the central nervous system (CNS), while others attribute them to systemic inflammation [7,8].

Studies supporting direct neurotropism include Beckman et al., who showed SARS-CoV-2 infection of neurons and associated neuroinflammation in a non-human primate model, and Song et al., who detected the virus in cortical neurons of both human and animal models. These findings suggest that SARS-CoV-2 can directly invade the CNS [8,9].

On the other hand, some researchers argue that neurological symptoms may stem from an exaggerated systemic immune response, rather than direct CNS infection. Pedrosa et al. reported that while SARS-CoV-2 infection in human neural cells is not permissive, it can trigger an inflammatory response, potentially causing indirect neural damage [10]. Similarly, Vanderheiden and Klein proposed that peripheral immune dysregulation and blood–brain barrier disruption could be driving factors in COVID-19 neuroinflammation [11].

The ongoing debate is further highlighted by Pröbstel and Schirmer, who summarized key evidence from both perspectives [12]. The case presented here adds to the discussion, by presenting a patient with confirmed SARS-CoV-2 neuroinfection, providing additional insight into the complexities of neurological involvement in COVID-19.

## 2. Materials and Methods

Health records were reviewed for details of clinical presentation and management. Standardized methods were used for molecular diagnostics [13]. Nasopharyngeal (NP) swabs in universal/viral transport media and clinical samples were collected from patients. Cerebrospinal fluid (CSF) was collected following standard recommendations using a 20-gauge needle [14]. The collected samples were tested immediately upon collection. Quantitative reverse transcriptase real-time PCR (qRT-PCR) testing was performed on clinical samples to determine a SARS-CoV-2 presence.

Viral RNA from nasopharyngeal swabs and cerebrospinal fluid (CSF) was extracted using the ExiPrep™ Dx Viral DNA/RNA Kit and the ExiPrep™ 16 Dx fully automated nucleic acid extraction system (BIONEER, Daejeon, Republic of Korea).

SARS-CoV-2 RNA detection via RT-PCR was performed with the LiliF™ SARS-CoV-2 Kit (iNtRON Biotechnology, Republic of Korea), which targets the E gene (HEX), N gene (ROX), and RdRp gene (FAM). RNase P (Cy5) was used as an internal control to verify RNA extraction from human-derived samples. A negative RNase P result, with positive findings for other targets, does not affect result interpretation.

The RT-PCR protocol was as follows: reverse transcription and Taq activation at 50 °C for 30 min, initial denaturation at 95 °C for 10 min, followed by 40 amplification cycles of 15 s at 95 °C and 60 s at 58 °C, using the Gentier Real-Time PCR System (Tianlong, Xi’an, China). A cycle threshold (Ct) value of ≤35 was considered positive, while values >35 were interpreted as negative.

The study adhered to the ethical standards outlined in the Helsinki Declaration, the International Ethical Guidelines for Health-related Research Involving Humans (2016), and the Personal Data Protection Law No. 25326. Consent was obtained for additional testing of clinically collected samples through Research Ethics Board No. 1481/13.

## 3. Results

### 3.1. Case Presentation

#### 3.1.1. Date of Admission: 14 September 2024

A 92-year-old cisgender female, with her only reported comorbidity being atrophic gastritis, was admitted to the hospital with a clinical picture characterized by progressive fatigue over the preceding several days. She presented with symptoms consistent with catarrhal syndrome, including a persistent cough, nasal congestion, and fever, recorded at 38.6 °C. Initial laboratory investigations revealed significant hypoxemia, with an oxygen saturation of 79.8%, along with metabolic acidosis indicated by a pH of 7.3 on arterial blood gas analysis (ABG). Inflammatory markers were notable for an elevated C-reactive protein (CRP) level of 14.8 mg/L and a markedly increased D-dimer level at 2.15 µg/mL, raising concerns for potential thromboembolic events.

Nasopharyngeal (NP) antigen testing and quantitative reverse transcription polymerase chain reaction (RT-PCR) confirmed the diagnosis of SARS-CoV-2 infection (Figure 1A and Table 1). Additionally, a nasopharyngeal swab identified a high bacterial load of penicillin- and ampicillin-resistant Staphylococcus aureus (10^5^ CFU/mL), necessitating careful consideration of antibiotic therapy. Chest X-ray failed to demonstrate any pathological changes (Figure 2). The patient was admitted for comprehensive management, with an initial treatment regimen comprising conservative measures, including 5 L per minute of oxygen supplementation, intravenous crystalloids for hydration, and subcutaneous fondaparinux for thromboembolic prophylaxis.

#### 3.1.2. Neurological Deterioration: 16 September 2024

Two days into her hospital stay, the patient exhibited the onset of neurological symptoms characterized by lethargy, discoordination, and impaired communication, notably without signs of meningism. A lumbar puncture was performed to obtain cerebrospinal fluid (CSF) for analysis, revealing the presence of SARS-CoV-2, as confirmed by RT-PCR with a cycle threshold (Ct) of <35 (Figure 1B and Table 1). Despite ongoing respiratory support, mild hypoxemia persisted, with an arterial oxygen saturation of 88.5%. Laboratory results indicated a further increase in CRP to 101 mg/L, suggesting heightened systemic inflammation. Chest X-ray findings revealed pneumofibrosis, indicating potential pulmonary complications. Consequently, the treatment regimen was escalated to include a daily administration of 10% mannitol for cerebral edema, dexamethasone 8 mg daily for inflammation, and ceftriaxone 2 g twice daily to address the bacterial infection.

#### 3.1.3. Clinical Course: 20 September 2024

On 20 September 2024, CSF microbiology results returned negative for bacterial pathogens, indicating that the observed neurological symptoms were not due to a secondary bacterial infection. However, despite the intensified treatment, there was no noticeable improvement in the patient’s neurological status. She remained somnolent, with a Glasgow Coma Scale (GCS) score of 10, raising concerns about her neurological prognosis. ABGs and electrolyte levels were monitored routinely and did not show significant deviations.

#### 3.1.4. Discharge Summary: 27 September 2024

After a 13-day hospital stay, the patient was discharged on 27 September 2024, with an improved GCS score of 14. However, she continued to experience persistent lethargy and cognitive difficulties, commonly described as “brain fog.” Inflammatory markers remained elevated, with a CRP level of 23 mg/L. Repeat RT-PCR analysis of the CSF confirmed an ongoing SARS-CoV-2 presence, while NP samples showed marginal positivity (Figure 1C,D and Table 1). These findings underscore the need for the continued monitoring and management of her neurological involvement. Table 2 summarizes the case’s key findings.

## 4. Discussion

This case report details persistent SARS-CoV-2 infection in the central nervous system (CNS) of a 92-year-old female patient, underscoring the neurotropism of SARS-CoV-2 and its potential to cause enduring neurological symptoms [15,16]. Although COVID-19 typically manifests with respiratory symptoms, substantial evidence indicates that SARS-CoV-2 can also impact the CNS, leading to both acute and persistent neurological manifestations [17]. The patient’s cognitive and motor decline, coupled with a negative CSF microbiological culture, suggests direct viral involvement, aligning with emerging evidence of variant-specific neurovirulence. Age-related neurodegeneration and immune decline likely increased her susceptibility to CNS complications. While SARS-CoV-2-related CNS infection remains a rare clinical occurrence, its potential for causing severe and persistent neurological impairments warrants attention [18]. Documented cases are limited, particularly in elderly patients, making this report a valuable addition to the understanding of SARS-CoV-2′s neurotropic behavior and its implications for geriatric care. COVID-19-related neurological symptoms can range from mild cognitive deficits to severe impairments, with symptoms like fatigue, memory loss, disorientation, and language difficulty. Frequently referred to as “brain fog,” these symptoms are commonly observed in long COVID cases, where cognitive and neurological symptoms persist beyond the acute infection phase. Recent studies further highlight a spectrum of neurological sequelae in COVID-19 patients, including stroke, encephalopathies, microbleeds, and inflammatory syndromes [19]. While direct CNS infection by SARS-CoV-2 is relatively rare, these cases highlight the virus’s capacity to induce significant neurological impairment, contributing to chronic CNS symptoms [20].

The neurotropic potential of SARS-CoV-2 may be partly explained by the expression of its entry receptor, ACE2, in CNS cells, such as neurons and glial cells. Although ACE2 expression is limited in axons and dendrites, the neuronal cell body shows higher levels, potentially allowing for SARS-CoV-2 entry. Previous SARS-CoV studies support this neuroinvasive potential, as SARS-CoV has been identified in the neurons of infected patients’ brains, suggesting that SARS-CoV-2 may behave similarly, due to its genetic similarity to SARS-CoV [21].

Neurological complications in COVID-19 patients appear to be both prevalent and enduring. For instance, a large cohort study of 236,379 COVID-19 patients reported that 33.6% exhibited neurological or psychiatric symptoms six months post infection, with rates reaching 46.4% in those admitted to intensive care [22]. These findings underscore SARS-CoV-2’s potential to impact neurological health long after acute illness, particularly in ICU patients who seem more vulnerable to lasting neurological complications.

The CNS involvement of SARS-CoV-2 can also vary according to viral variants. Research by Proust et al. demonstrated that SARS-CoV-2 variants have distinct impacts on CNS cells and the blood–brain barrier [23]. While the original strain showed broad cytopathic effects on CNS cells, the Alpha and Beta variants specifically affected pericytes, and the Omicron variant targeted endothelial cells and pericytes. Such findings imply that each SARS-CoV-2 variant may exhibit unique neurovirulence characteristics, influencing its potential for CNS involvement [24].

In the presented geriatric case, neurological symptoms, including lethargy, poor coordination, and impaired communication, developed without meningeal signs. CSF analysis confirmed the presence of SARS-CoV-2 RNA, while bacterial cultures were negative. The repeated detection of SARS-CoV-2 in the CSF suggests the potential for long-term CNS infection, which likely contributed to the patient’s ongoing neurological symptoms. While an MRI was not performed, studies have noted the absence of specific MRI findings in SARS-CoV-2-positive CNS cases, indicating that MRI may not consistently detect COVID-related CNS involvement [25].

Patient age is a significant factor in the CNS impact of SARS-CoV-2 infection. Pediatric cases show variable rates of CNS manifestations across different pandemic waves, with higher rates during the Wuhan and Omicron waves, and younger children, particularly those with a median age of 3.2 years, being more frequently affected by CNS symptoms such as seizures and headaches [26]. In contrast, older adults experience more severe neurological symptoms, including stroke, encephalitis, and myelitis, likely due to age-related changes in the brain and immune system [27,28].

This case report has several limitations, especially regarding long-term follow-up. In Bulgaria, there is a lack of structured national follow-up and recovery programs for COVID-19 patients [29]. Without these programs, tracking long-term outcomes and recovery trajectories becomes challenging, particularly for elderly patients who may have limited access to specialized care. Furthermore, the absence of comprehensive, centralized databases restricts our ability to monitor persistent symptoms and complications, potentially leading to the underreporting of long-COVID and neuro-COVID cases [30]. Additionally, the lack of routine viral genome sequencing limits the ability to confirm specific SARS-CoV-2 variants responsible for CNS involvement. While qRT-PCR reliably detects viral RNA, genome sequencing could provide critical insights into variant-specific neurovirulence and mutation profiles linked to persistent CNS infections [31]. Implementing sequencing in clinical practice could enhance diagnostic accuracy, guide tailored treatments, and improve understanding of variant-associated clinical outcomes.

These limitations emphasize the need for dedicated follow-up initiatives to offer ongoing support, collect critical data on chronic symptoms, and refine our understanding of COVID-19’s long-term impact on neurological health. Establishing a systematic follow-up protocol could provide invaluable insights into disease progression and recovery, supporting the development of more targeted treatment approaches for patients with CNS involvement.

## 5. Conclusions

This case highlights the complexity of CNS involvement in COVID-19, and underscores the need for ongoing research to understand SARS-CoV-2 neurotropism and neurovirulence. Although COVID-19 severity does not always correlate directly with increased CNS involvement, our findings indicate that SARS-CoV-2 can persist in the CNS, causing prolonged neurological impairment. Further studies on variant-specific CNS effects and pathophysiology are essential to develop effective management strategies for COVID-19 neuroinfections and long COVID.

## Figures and Tables

**Figure 1 viruses-16-01962-f001:**
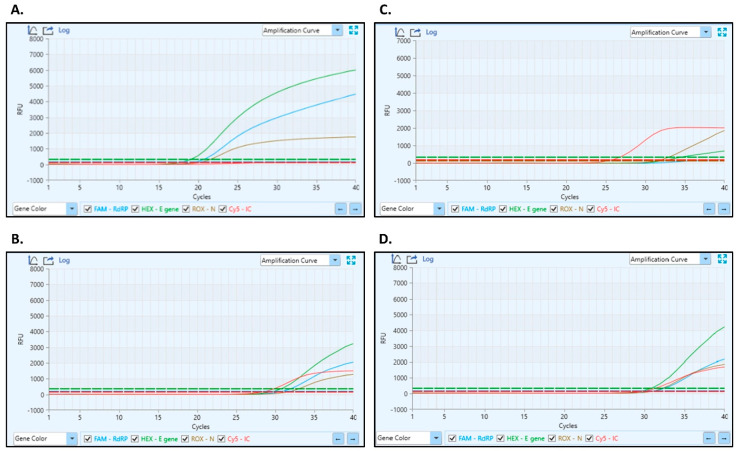
Cycle threshold (Ct) values for target genes in nasopharyngeal (NP) swabs and cerebrospinal fluid (CSF) samples at initial presentation and discharge. (**A**) Initial NP Swab Ct Values: RdRp = 19.910, E gene = 19.168, and N gene = 20.105. (**B**) Initial CSF Ct Values: RdRp = 29.074, E gene = 28.121, and N gene = 28.449. (**C**) NP Swab Ct Values at Discharge: E gene = 34.500 and N gene = 31.246. (**D**) CSF Ct Values at Discharge: RdRp = 31.900, E gene = 31.000, and N gene = 30.949.

**Figure 2 viruses-16-01962-f002:**
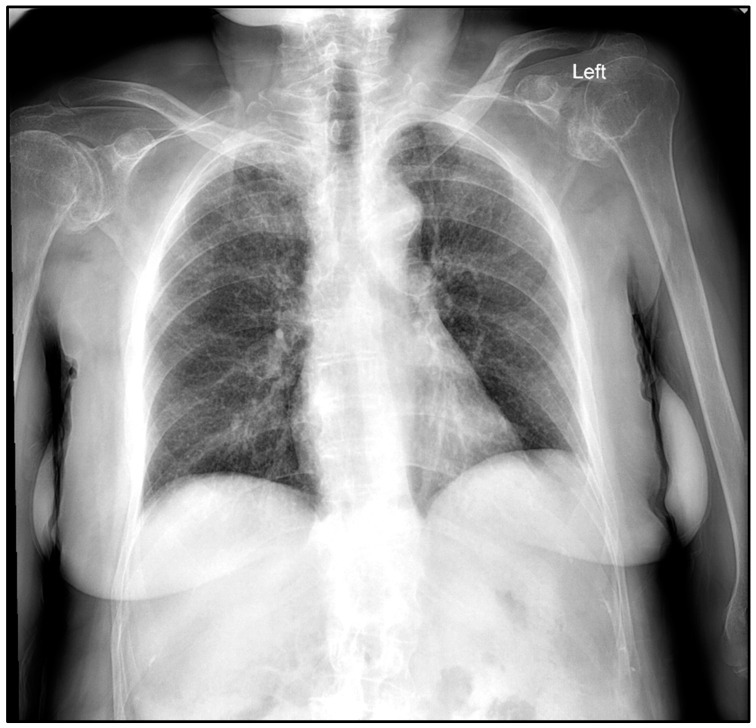
Chest X-ray at initial presentation.

**Table 1 viruses-16-01962-t001:** Summary of RT-PCR sample.

Specimen Type	qRT-PCR Cycle Threshold (Ct) Value	Assay Target (s)
NP Swab initial **	RdRp = 19.910	E gene = 19.168; N gene = 20.105
NP Swab discharge **	ND	E gene = 34.500; N gene = 31.246
CSF initial *	RdRp = 29.074	E gene = 28.121; N gene = 28.449
CSF discharge *	RdRp = 31.900	E gene = 31.000; N gene = 30.949

** RNA was extracted from 140 µL of sample. * RNA was extracted from 28 µL of sample. CSF-cerebrospinal fluid, NP—nasopharyngeal, ND—not detected; E gene Ct values determined by the LiliF™ SARS-CoV-2 assay.

**Table 2 viruses-16-01962-t002:** Summary of key findings.

Category	Findings
Patient Demographics	92-year-old female with moderate COVID-19 symptoms
Imaging	No pathological findings on chest X-ray
Neurological Symptoms	Lethargy, poor coordination, impaired communication on day 3 post submission—without signs of meningism
Blood Tests	Initial laboratory findings revealed hypoxemia (O_2_ saturation 79.8%), acidosis (pH 7.3), an elevated CRP level of 14.8 mg/L, and a high D-dimer level (2.15 µg/mL)
CSF Analysis	SARS-CoV-2 RNA detected by qRT-PCR on day 3 (Ct = 29) after hospitalization and at discharge (day 13 with Ct = 31); negative microbiological culture
Clinical Course	Persistent neurological deficit despite standard COVID-19 treatment, even after discharge
Key Implications	Evidence of CNS viral RNA persistence

CRP—C-reactive protein, RNA—ribonucleic acid, CNS—central nervous system, qRT-PCR—real-time polymerase chain reaction.

## Data Availability

The dataset presented in this article was obtained during routine clinical patient care and is not readily available because of patient privacy protection.

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
