# Peer review of "SARS-CoV-2 Infection of the Central Nervous System: A Case Report"

_viruses, 2024, doi:10.3390/v16121962_

Round 1
Reviewer 1 Report
Comments and Suggestions for Authors
The manuscript by Valkov et al describes a case repot of SARS-CoV-2 infection of the central nervous system. This case report describes the clinical progression of a 92-year-old female who developed a persistent neuroinfection associated with SARS-CoV-2. The manuscript is a simple case study based on one patient. The authors make case for publication that the information could add to the ongoing discussion of SARS-CoV-2 transmission to CNS. Given the limitation of sample collections two points need to be addressed.
1. The authors need to expand and distantly explain the new information in the current case report that warrant publication compared to previous reports. Given the fact that this a one case report and without statistical significance lowers the impact.
2. Does CSF collection from lumber and identification by qRT-PCR alone could lead to neuroinfection conclusion the author reached? What are the limitation and the weakness of such conclusion?
Author Response
Reviewer 1:
- The authors need to expand and distantly explain the new information in the current case report that warrant publication compared to previous reports. Given the fact that this a one case report and without statistical significance lowers the impact.
We appreciate this valuable feedback. While single-patient case reports inherently have limited statistical power, our case highlights important novel aspects of SARS-CoV-2 CNS infection that contribute to the broader understanding of the disease:
- Persistent CNS Involvement: Our patient demonstrated prolonged SARS-CoV-2 RNA detection in the cerebrospinal fluid (CSF) without bacterial coinfection. This supports the possibility of chronic CNS infection even in geriatric populations, a phenomenon not frequently documented.
- Variant-Specific Neurotropism: Our discussion addresses the evolving understanding of SARS-CoV-2 variants' differential effects on CNS cells, supported by emerging literature.
- Clinical Implications and Public Health Challenges: We underscore the lack of follow-up programs in Bulgaria, emphasizing the need for more robust monitoring of post-COVID neurological sequelae.
Implementation in manuscript:
We expanded the Discussion section by adding a detailed paragraph summarizing the novel aspects of the case report, emphasizing persistent SARS-CoV-2 CNS infection, variant-specific neurovirulence, and the clinical implications for geriatric patients with limited follow-up options.
- We acknowledge that using qRT-PCR from lumbar CSF samples alone has inherent limitations when concluding SARS-CoV-2 neuroinfection:
- Limited Diagnostic Scope: qRT-PCR detects viral RNA but does not confirm active viral replication or direct CNS inflammation, limiting its ability to establish a definitive neuroinfection diagnosis.
- False Positives and Contamination Risks: While repeated CSF testing reduced the likelihood of false positives in our case, the possibility of contamination during sample handling cannot be entirely ruled out.
- Absence of Supporting Biomarkers: Additional diagnostic tests such as cytokine profiling, neuroinflammatory markers, or histopathological examination were unavailable, restricting a more comprehensive assessment of CNS involvement.
Despite these limitations, qRT-PCR remains the gold standard for detecting specific viral genetic material, offering valuable diagnostic insight when interpreted alongside clinical findings and imaging results. In vitro sample cell inoculation, while useful for confirming active viral replication, is not routinely performed in clinical practice due to its complexity and limited availability.
Reviewer 2 Report
Comments and Suggestions for Authors
1. Suggest that the authors reorganize the Results section. Using a table to summarize key findings may help readers follow the content more easily.
2. Since q-PCR is highly sensitive, to obtain more reliable data, it is better to acquire sequence information of the SARS-CoV-2 genome (as long as possible) from the CSF samples at both initial and discharge stages."
3. All the citations should be double-checked to ensure that the references are cited properly. For example, in line 47: reference (7), the paper has been published in the Journal of Virology (PMID: 37039676), but the authors still cited the preprint version.
4. Carefully review the paper and correct any minor mistakes, such as in the figure legend of Figure 1, where 'A' is bold, but 'B,' 'C,' and 'D' are not.
Author Response
Reviewer 2:
- Suggest that the authors reorganize the Results section. Using a table to summarize key findings may help readers follow the content more easily.
Thank you for this helpful suggestion. Added Table 2 summarizing the case's key findings.
- Since q-PCR is highly sensitive, to obtain more reliable data, it is better to acquire sequence information of the SARS-CoV-2 genome (as long as possible) from the CSF samples at both initial and discharge stages.
While we agree that viral genome sequencing would strengthen our findings, obtaining these data was not feasible during the patient's care due to logistical constraints as we do not perform this analysis routinely. We will acknowledge this limitation in the Discussion section and recommend sequencing in future studies for a more definitive diagnosis.
- All the citations should be double-checked to ensure that the references are cited properly. For example, in line 47: reference (7), the paper has been published in the Journal of Virology (PMID: 37039676), but the authors still cited the preprint version.
We appreciate this important note. We have reviewed all references carefully and corrected any outdated preprint citations, including Reference 7. Used EndNote during this process.
- Carefully review the paper and correct any minor mistakes, such as in the figure legend of Figure 1, where 'A' is bold, but 'B,' 'C,' and 'D' are not.
We revised the manuscript for consistency as per the reviewer’s recommendation.
Round 2
Reviewer 1 Report
Comments and Suggestions for Authors I believe the authors addressed my criticisms and I suggest the revised manuscript to be accepted.Reviewer 2 Report
Comments and Suggestions for Authors
No further comments